# Long-Term Follow-Up of Thyroid Incidentalomas Visualized with 18F-Fluorodeoxyglucose Positron Emission Tomography—Impact of Thyroid Scintigraphy in the Diagnostic Work-Up

**DOI:** 10.3390/diagnostics11030557

**Published:** 2021-03-19

**Authors:** Kirsten Korsholm, Michala Reichkendler, Louise Alslev, Åse Krogh Rasmussen, Peter Oturai

**Affiliations:** 1Department of Clinical Physiology, Nuclear Medicine and PET, Copenhagen University Hospital, Rigshospitalet, DK-2100 Copenhagen, Denmark; michala.reichkendler.01@regionh.dk (M.R.); alslev_louise@hotmail.com (L.A.); Peter.Sandor.Oturai@regionh.dk (P.O.); 2Department of Endocrinology, Copenhagen University Hospital, Rigshospitalet, DK-2100 Copenhagen, Denmark; Aase.Krogh.Rasmussen@regionh.dk

**Keywords:** thyroid, incidentalomas, thyroid malignancy, thyroid scintigraphy, 18F-FDG-PET

## Abstract

Our objective was to evaluate the frequency of malignancy in incidental thyroidal uptake on 18F-fluorodeoxyglucose positron emission tomography/computed tomography (18F-FDG PET/CT) in a cohort of Danish patients, and furthermore to evaluate the impact of thyroid scinti-graphy in the diagnostic work-up. All whole-body PET/CT reports from 1 January 2010 to 31 December 2013 were retrospectively reviewed and further analyzed if visually increased thyroidal FDG uptake was reported. Patient electronic files were searched for further thyroid evaluation. Of 13,195 18F-FDG-PET/CT scans in 9114 patients, 312 PET/CT reports mentioned incidental thyroid FDG-uptake, and 279 patients were included in the study (3.1%). The thyroid was further investigated in 137 patients (49%), and 75 patients underwent thyroid scintigraphy. A total of 57 patients had a thyroid biopsy and 21 proceeded to surgery. Surgical specimens displayed malignancy in 10 cases, and one thyroid malignancy was found by autopsy. Hence, 11 patients were diagnosed with thyroid malignancies among 279 patients with incidental thyroid 18F-FDG uptake (3.9%). In 34 patients, a biopsy was avoided due to the results of the thyroid scintigraphy. We conclude that patients with thyroid incidentalomas can benefit from further diagnostic work-up including a thyroid scintigraphy.

## 1. Introduction

Incidental thyroid nodules (ITN) are defined as nodules originally detected on ima-ging studies performed in patients with no thyroid-related clinical symptoms, examination findings, nor suspicion of thyroid disease [1]. Due to the increased use of imaging, more and more ITN are discovered, as thyroid nodules are common in the general Danish population.

Population studies in Denmark have shown that 15% of women in the age group 40 to 45 years have multiple nodules >1 cm in their thyroid gland, and that the prevalence increases to 25–30% in women age 60–65 years [2]; however, in Denmark the incidence of thyroid malignancy is only 360 new cases per year (6.2 cases per 100,000 inhabitants/year) (https://www.dahanca.dk/assets/files/Aar_r_DAHANCA_2018.pdf, accessed on 26 June 2020). In the United States, the incidence of thyroid cancer nearly tripled from 1975 to 2009, and despite this change in incidence, the mortality rate from thyroid cancer remained stable [3]. This is partly explained by over-diagnosing and partly by the fact that small thyroid cancers can have an indolent course, and an ideal approach would be only to diagnose those cancers that are likely to be of clinical significance.

18F-Fluorodeoxyglucose positron emission tomography (18F-FDG-PET) visualizes cells with increased glucose demand such as malignantly transformed cells; however, this high glucose uptake is nonspecific and may also represent inflammation, infection or increased glandular activity [4,5].

Large studies and systematic reviews report a prevalence of 1−2.5% of ITNs with increased FDG-uptake in PET scans, with a frequency of malignancy of the ITNs with focal FDG uptake of approximately 33% [6,7,8,9]. To discriminate between benign and malignant thyroid tissue, researchers have thoroughly investigated standardized maximum FDG uptake value (SUVmax) of the ITNs. A large meta-analysis of 16 studies [9] reported that 9 studies demonstrated a significant difference between malignant and benign lesions, while the other 7 found no such difference, indicating that there is no safe cut-off value below which malignancy can be ruled out.

The American Thyroid Association recommends that patients with focal metabolic activity in the thyroid detected on 18F-FDG-PET scans undergo dedicated thyroid ultrasonography and fine-needle aspiration cytology (FNAC) if there is a solid lesion ≥1 cm corresponding to the FDG-avid lesion [10]. However, unless the cytology result is definitively benign or malignant, patients may undergo repeated biopsies or diagnostic surgery. Hoang et al. reported that 25−41% of patients who undergo FNAC for ITNs will proceed to surgery, and 36−75% of these patients will have benign nodules [11]. The context of underlying disease and life expectancy should therefore be kept in mind when further investigations are initiated [12].

In Denmark, a thyroid scintigraphy is a common part of the evaluation of thyroid nodules, and a Danish study suggested that patients with ITN detected by 18F-FDG-PET should undergo thyroid scintigraphy and ultrasonography followed by FNAC if a nodule is hypo-functioning [13].

In our center, approximately 6000 whole-body 18F-FDG-PET scans are performed yearly (2016–2019). If an FDG-positive ITN is detected, the patient may be referred for further diagnostic procedures, usually including a thyroid scintigraphy and ultrasonography. We find it of interest to investigate the frequency of FDG-avid ITNs in Eastern Denmark, the frequency of malignant lesions, and the thyroid scintigraphy findings in patients with FDG-avid lesions in the thyroid. As scintigraphically hyperfunctioning nodules have a very low risk of malignancy [14], unnecessary biopsies and surgery might be prevented when using a thyroid scintigraphy in the diagnostic work-up. To our knowledge, only one other study has addressed the combined use of thyroid scintigraphy and FDG-PET to detect possible malignant lesions in the thyroid [15], however, no study has investigated the impact of thyroid scintigraphy in the diagnostic work-up of FDG-avid ITNs in a large cohort of patients.

## 2. Materials and Methods

### 2.1. Patients

From 1 January 2010 to 31 December 2013, 13,195 whole-body 18F-FDG-PET/CT scans were performed in the Department of Clinical Physiology, Nuclear Medicine, and PET, Copenhagen University Hospital, Rigshospitalet, in 9114 patients. Of these, 312 scans reported increased FDG uptake in the thyroid gland. All 312 scan reports were individually reviewed, resulting in 279 patients included in the study. Patients were excluded if they had known or suspected malignant thyroid disease, or if they had a previous 18F-FDG-PET/CT scan, where increased thyroid FDG uptake was reported. In addition, 18F-FDG-PET/CT scans performed for radiation therapy planning or scans performed in research protocols were excluded. All medical files were searched, and the clinical decisions upon the results of the 18F-FDG-PET/CT report were scrutinized. Data concerning thyroid scintigraphy, thyroid ultrasonography, plasma-stimulating thyroid hormone (P-TSH), FNACs, coarse needle biopsies, pathological examination of surgical specimens, clinical follow-up, and date of eventual death were sampled. In addition, all patients were searched in The Danish Pathology Data Bank (a database containing results of all Danish pathological examinations).

The main part of the patients came from Eastern Denmark.

### 2.2. PET Imaging

The PET/CT images were obtained using Biograph mCT-S (64–128 slices) and Biograph TrueV (40–64 slices) scanners (Siemens Medical Solutions, Malvern, PA, USA) and a Discovery LS (16 slices) scanner (GE Healthcare, Waukesha, WI, USA). The patients were intravenously injected with 4 MBq/kg 18F-FDG after a minimum of 6 hours of fasting, and the PET/CT was performed approximately 60 min after tracer injection. The CT scan was performed as diagnostic CT with contrast-enhancement or low-dose CT without contrast.

Images were originally interpreted by a team consisting of a nuclear medicine physician and a radiologist, and pathological FDG-uptake was reported when detected. Subsequently, 18F-FDG-PET scans of patients included in the study were re-evaluated, and pathological FDG uptake in the thyroid was classified as either focal, multifocal, or diffuse; maximum standardized uptake value (SUVmax) was obtained for the FDG-avid areas. All the ITNs were mentioned in the original report from the 18F-FDG-PET scan, however, not all ITNs were mentioned in the conclusion of the report, e.g., if the patient was critically ill of disseminated cancer, the radiologist and nuclear medicine physician decided not to highlight the ITN as part of the concluding remarks.

### 2.3. Thyroid Scintigraphy

Following the administration 100–200 MBq 99mTc-pertechnetate, thyroid scintigraphy was performed on a single-head gamma-camera using high-resolution parallel-hole and pinhole collimators (Mediso, Budapest, Hungary). Image matrix was 256 × 256, acquisition time 10 min, and no SPECT/CT was performed. In our department, 99mTc-pertechnetate was the agent of choice due to high-quality images, low cost, and daily availability.

### 2.4. Blood Tests

The main part of the patients underwent blood tests; however, it was not undertaken consequently. P-TSH was the first step in the blood analyses (reported in Table 1); data from within a timespan of ±30 days within index PET/CT were recorded.

### 2.5. Cyto/Histopathology

The FNACs were categorized according to The Bethesda System for Reporting Thyroid Cytopathology (TBSRTC) introduced in 2007 to standardize terminology used in reporting thyroid cytology [16]. When histopathology was available, results were compiled as malignant or benign.

### 2.6. Statistics

All statistical tests were performed using SPSS ver. 25 (SPSS IBM, Armonk, NY, USA). Test of normality was calculated using the Shapiro–Wilk test. Differences in age between groups were calculated using Students *t*-test. Difference in proportion of deaths between groups was calculated with Pearson’s chi-squared test. Difference between FDG SUVmax of patients with thyroid malignancy vs. SUVmax of patients with no proven thyroid malignancy was calculated using Mann–Whitney test and Kruskal–Wallis test. *p*-values less than 0.05 were considered significant. Kaplan–Meier curves were used to illustrate survival in the group receiving further thyroid evaluation vs. the group who did not receive further thyroid evaluation. In addition, a log-rank test was used to test differences in mortality between the 2 groups.

### 2.7. Approvals

This study was approved by the Danish Patient Safety Authority (no. 3-3013-1528/1/) (25 June 2016), in addition to the Danish Data Protection Agency at Region Hovedstaden (RH-2015-227, I-Suite 04178) on 27 October 2015. As this was a retrospective study, no approval from the regional Ethical Committee was acquired according to national legislation.

## 3. Results

### 3.1. Patients

A total of 279 patients with increased 18F-FDG-uptake in the thyroid (279/9,114 = 3.1%) were identified (Figure 1). The patients were referred to 18F-FDG-PET/CT either because of suspicion of cancer, monitoring of cancer treatment, suspicion of recurrence of cancer, or to characterize various non-malignant diseases.

A total of 216 (77%) were women, and 63 (23%) were men. Average age was 61 years (range 26–88 years) at the time of index PET/CT. At data cut-off, June 2020, 136 (49%) of the 277 patients had deceased. For two patients, we do not have information on eventual death/survival.

Median length of follow-up period for the deceased was 19.5 months (range 0–118 months); for those still alive at June 2020, median length of follow-up period was 99 months (range 77–124 months).

In 137 patients (49%), the referring department/physician decided to further examine the thyroid gland, and in 140 patients, no further thyroid-related examinations were performed; two patients were lost to follow-up. The patients in the group with further thyroid evaluation were significantly younger than those with no further thyroid evaluation (Table 1). Fifty-six patients underwent screening of P-TSH within ±30 days of the index PET/CT (Table 1).

### 3.2. Thyroid FDG Uptake

Of the 279 patients with increased thyroid FDG uptake, 160 represented focal uptake (57%), 50 were multifocal (18%), and 69 were diffuse (25%). When excluding the diffuse high uptake, the frequency of FDG-avid nodules was 2.3% (210/9,114).

### 3.3. Imaging and Pathology in the Diagnostic Work-Up

Among the 137 patients with further thyroid evaluation, 71 patients (52%) had ultrasonography performed in combination with thyroid scintigraphy, 20 (15%) had only ultrasonography, and 4 (3%) patients had only thyroid scintigraphy performed (Figure 1).

Among the 75 patients with a thyroid scintigraphy, 34 had normal-/hyper-functioning tissue/nodules, and in this way biopsy was avoided (if a nodule was unambiguously scintigraphically hyperfunctioning, no FNAC was performed).

Forty-one patients had either hypofunctioning nodules/tissue or were inconclusive (*n* = 1), and in one case, information was missing (Table 1). Thirty-five patients (of the 41 with hypofunctioning nodules/tissue) were referred for FNAC (4 of the 41 patients had diffuse low uptake scintigraphically and were not referred for FNAC, and in two cases, FNAC was not performed for unknown reasons).

Overall, 57 patients had either FNAC (*n* = 52) or coarse needle biopsy (*n* = 5) from the thyroid. Of these patients, 35 had a preceding thyroid scintigraphy and ultrasonography examination, 18 only had ultrasonography, and 4 cases had no information about thyroid scintigraphy or ultrasonography-guided biopsies (Figure 1). The biopsies showed malignant cells in nine cases (Bethesda VI), suspicion of malignancy in one case (Bethesda V), follicular neoplasia in seven cases (Bethesda IV), and atypical changes in two cases (Bethesda III); moreover, eight biopsies were not representative/inconclusive (Bethesda I), and there were no signs of malignancy in 30 biopsies (Bethesda II). Nine patients had thyroid surgery due to Bethesda category VI, one due to Bethesda V, three due to Bethesda IV, two due to Bethesda III, and two due to Bethesda I. Four patients underwent thyroid surgery despite lack of malignant/suspicious cytology (see below).

In total, 21 patients had either hemi- or total thyroidectomy performed (Figure 1).

In 10 patients, thyroid surgery revealed malignant histopathology (eight with cytology Bethesda VI, one with Bethesda V, and one with Bethesda I). In one patient with cytology Bethesda VI, thyroid surgery revealed a follicular adenoma. Among the patients with malignant thyroid histopathology, in detail, we found five cases with papillary adenocarcinoma, one case with two synchronous cancers in the same lobe (a papillary adenocarcinoma and a medullary carcinoma), one case with low-differentiated adenocarcinoma, one with follicular carcinoma, one squamous cell carcinoma from the thyroid, and one with diffuse large cell B cell lymphoma. Additionally, autopsy in one patient revealed a metastasis from a renal cell carcinoma in the thyroid.

Biopsies were categorized as other than Bethesda II in 27 patients (Figure 1). Of these, 17 underwent thyroid surgery, and 10 did not—often due to patients’ own wish. Instead, eight of these were controlled with thyroid ultrasound and/or thyroid scintigraphy—some patients for up to six years with no measurable growth on ultrasound. One patient denied surgery or further follow-up, and in one case information on the further course was missing. These numbers are displayed graphically in Figure 1.

In conclusion, we found pathology proved malignancy of the thyroid in 11 patients in our cohort of 279 patients, giving a proportion of 3.9% of all FDG-avid thyroid lesions. Among the 137 patients who received further thyroid evaluation, the proportion was 8.0%, and among all with surgical specimens/autopsy (*n* = 22), a proportion of 50% (Table 1). A search in the Danish Pathology Data Bank of all patients at the end of our follow-up period revealed no further malignant thyroid disease than those described above.

Of the 11 patients with a proven malignancy in the thyroid, 6 had focal high FDG-uptake in the thyroid, 4 had multifocal high FDG uptake, and 1 had diffuse high thyroid uptake (the latter patient was diagnosed with diffuse large cell B cell lymphoma in the thyroid).

In 30 biopsies, there were no signs of malignancy (Bethesda II)—of these, 24 demonstrated normal thyroid tissue, 3 were colloidal goiter, 1 nodular goiter, 1 oncocytoma, and 1 lymphocyte infiltration.

In four patients, it was decided to perform a total or a hemi-thyroidectomy despite lack of malignant cytology: one case due to paresis of the recurrent laryngeal nerve, one case according to the patient’s own wish, one case due to severe pain in the thyroid, and one case due to strong suspicion of a metastasis to a cervical lymph node very close to the thyroid in a patient with cervical cancer; however, all surgical specimens from this group demonstrated no signs of malignancy.

### 3.4. Thyroid Scintigraphy and Pathology Outcome

Of the 11 patients with pathology-proven malignancy in the thyroid, 5 had a preceding thyroid scintigraphy. All five thyroid scintigraphs displayed hypofunctioning nodules (Figure 2a,b and Table 1). Among the 39 with scintigraphically cold nodules/tissue, 5 (12.8%) were malignant.

Among all patients with scintigraphically hyperfunctioning nodules, when searching The Danish Pathology Data Bank in June 2020, we found no subsequent thyroid malignancies.

### 3.5. FDG SUVmax and Pathology

SUVmax in the group of patients with proven thyroid malignancy (*n* = 11) was significantly higher compared with patients with no proven malignancy (*n* = 125, one SUV-max value missing due to technical problems) (*p* = 0.006), with a substantial overlap in the range of SUVmax (Figure 3).

### 3.6. Survival

In the group with further thyroid evaluation, 51 patients (37%) died during follow-up; in the group with no further thyroid evaluation, 85 patients (61%) died during the follow-up period (*p* < 0.0001).

Until data cut-off in June 2020, the median survival time after index 18F-FDG-PET/CT of all patients with increased thyroid uptake was 104 months. Median survival time in the group with no further thyroid evaluation was 44 months. In the group with further thyroid evaluation, median survival time was not reached within follow-up time, however, 75% were alive after 54 months. In patients with no further thyroid evaluation, 75% survival time was significantly shorter, namely 12 months (*p* < 0.0001; Figure 4). In the group lost to follow up (two patients), no information on eventual survival/death could be obtained.

## 4. Discussion

Nodules in the thyroid is a common phenomenon, especially in countries with known iodine deficiency. Denmark was previously an iodine-deficient country, however, since mandatory iodine fortification of household- and bread salt was introduced in 2000, Eastern Denmark is today iodine-sufficient.

In our study, we found incidentally high FDG uptake in the thyroid in 3.1% of all 18F-FDG-PET/CT scans; when excluding diffuse high uptake, the frequency was 2.3%. This is in accordance with other studies, especially one other Danish study by Gedberg et al. [17], who reported a frequency of focal thyroid uptake of 2.4%. A large meta-analysis by Bertagna et al. from 2012 [9] reported that thyroid incidentalomas are relatively frequent, ranging from 0.2 to 8.9%, with a pooled incidence of 2.5%, and other large studies have reported detection rates between 1.0 and 2.3% [6,7,8,18], also in correspondence with our results.

Diagnostic work-up of thyroid incidentalomas is important; however, this aspect is strongly associated with primary disease status of the patient and eventually limited expectancy of life. In our study, 137/279 patients (49%) received further thyroid evaluation. Many of our patients were diagnosed with a non-thyroid cancer in an advanced stage, and the decision not to commence further evaluation of the thyroid could be due to limited life expectancy. In other studies, the percentage of patients with ITNs referred for further thyroid evaluation ranges from 29% to 58% [6,7,17,18]; in detail, in a systematic review, Soelberg et al. [6] reported focal FDG uptake in the thyroid in 1.6% (1994 patients) of 125,754 individuals, wherein 1051 patients (53%) received further thyroid evaluation of which 35% had thyroid malignancy. Shie et al. [7] included 55,160 patients, of which 571 patients had focal FDG uptake in the thyroid (1%), 332 patients (58%) received further thyroid evaluation, and 33.2% of these demonstrated thyroid malignancy.

The frequency of malignancy in the thyroid in our study was 11 patients of 279 patients included, that is, 3.9%; however, if we only take into consideration the patients who received further thyroid evaluation, the frequency is 11/137 = 8.0%, and among all with surgery/autopsy, the frequency is 11/22 = 50%. In the literature, different frequencies are reported, and it is strongly recommended that one closely study how these rates are calculated, as there is a large variation in the methods of reporting. Gedberg et al. [17] reported in a Danish cohort a malignancy rate among all patients with focal FDG uptake in ITNs of 16.9%; among all patients with further thyroid evaluation, 30.3%; and among all with conclusive pathology, 50%. The latter number is in complete accordance with our results. However, there is a tendency in larger analysis to report malignancy frequencies only among the patients with further thyroid evaluation, giving a rather high frequency, as many patients never receive further thyroid evaluation, e.g., in Bertagna et al.’s study from 2013 [18], 49,519 cases were evaluated retrospectively, 729 patients had an ITN, 211 patients (29%) received further thyroid evaluation, and 72 patients had a thyroid malignancy, giving a frequency of 72/211 = 34%; however, if we calculate the frequency differently: 72/729, it results in a malignancy frequency of 9.9%.

In our study, one patient with diffuse high thyroid FDG uptake was diagnosed with thyroid malignancy (diffuse large cell B cell lymphoma in the thyroid), while the others with diffuse high thyroid FDG uptake displayed non-malignant conditions. In the study by Soelberg et al. [6], diffuse uptake in the thyroid was seen in 2.1%, and the prevalence of malignancy among these patients was 4.2%, while the remaining patients had thyroiditis or other non-malignant disease; hence, diffuse FDG uptake rarely indicates thyroid malignancy, yet it cannot exclude it.

We chose to have a long follow-up period, up to 10 years (minimum 6 years), as many thyroid malignancies are slow-growing, and autopsies have revealed unknown nodules in the thyroid in 50–60% of the population [19]. However, there is one drawback of this choice—as our patients were included from 1 January 2010 to 31 December 2013, the ultrasonography examinations did not report high-risk features of the nodules according to EU-TIRADS, as this strategy was not implemented until 2017 [20]. A recent Danish study evaluated the use of ultrasonography in diagnostic work-up in patients with ITN, reporting that an ultrasonography feature of irregular margins of the nodule gives the highest sensitivity and specificity with regard to malignancy; however, the authors commented that this feature is rather subjective as there are no strict rules on what an irregular margin looks like, and therefore this definition may vary among examiners [21].

Survival time was significantly longer in the group with further thyroid work-up compared to the group with no further thyroid evaluation. Our interpretation is that patients who received further work-up were those with less advanced primary disease (e.g., non-thyroid cancer), and therefore eligible for further work-up, whilst those in the other group were in an advanced stage of their primary disease and probably not eligible for or willing to participate in any further work-up. A search in the National Pathology Data Bank did not reveal any further thyroid malignancies in this group, e.g., from autopsies, and we believe that the reason for the higher mortality in this group is their primary non-thyroid disease.

We observed FDG SUVmax values to be significantly higher in patients with thyroid malignancies compared to patients with no thyroid malignancies, however, with an overlap in SUVmax values between benign and malignant lesions; this is in accordance with the literature where, e.g., Thuillier et al. [22] reported higher SUVmax values in malignant lesions than in benign lesions, however, not significantly different. Highest SUVmax values were also found in malignant lesions in a study by Hagenimana [23], but again no SUVmax cut-off value to distinguish between benign and malignant lesions could be identified.

In Denmark, a thyroid scintigraphy is a common part of the diagnostic work-up of patients with palpable nodules in the thyroid, and in this way unnecessary biopsies may be prevented as scintigraphically normal- or hyper-functioning nodules very seldom are malignant [14]. However, there is one drawback—many patients receive iodine-containing contrast medium intravenously in relation to their PET/CT scan, and this excess inorganic free iodide may reduce thyroid uptake of 99mTc pertechnetate, potentially resulting in a non-diagnostic thyroid scintigraphy up to 8–12 weeks after the contrast-enhanced PET/CT-scan. Andersen et al. [24] addressed this issue and concluded that the image quality of a thyroid scintigraphy after contrast-enhanced CT is age-dependent, recommending the following delay from contrast-enhanced CT to thyroid scintigraphy: 4 weeks for patients aged younger than 50 years, 6 weeks for patients aged 50 to 60 years, and 8 weeks for patients older than 60 years.

In our study, 55% (75/137) had a thyroid scintigraphy as part of the thyroid evaluation, and 45% (34/75) of these had hyperfunctioning or normal-functioning thyroid nodule(s)/tissue. In this way, a FNAC was prevented in 34 patients.

Not all patients underwent a thyroid scintigraphy. In 62 patients (45%) of the 137 patients receiving further thyroid evaluation, a scintigraphy was not performed, and various (unreported) reasons explaining this may be (1) a diffuse high FDG uptake (*n* = 69) indicating a low risk of malignancy; (2) some patients received CT contrast agents that excluded scintigraphy the following 4–8 weeks, and the referring clinicians did not want to postpone further evaluation, and performed US and FNAC instead; (3) the oncologists/clinicians referring the patients to 18F-FDG-PET were unaware of the role of scintigraphy in the work-up of (unpalpable) 18F-FDG-PET-avid thyroid nodules.

We have only knowledge of one other study combining the results of thyroid scintigraphy and 18F-FDG-PET of the thyroid, namely, the study by Wolf et al. [15], which describes how 18F-FDG-PET can verify benignancy/malignancy of cold nodules in the thyroid displayed by thyroid scintigraphy. Their main findings agree with our results, although their approach to the subject is different from ours.

We observed a high predictive value of normal or increased tracer uptake in the thyroid scintigraphies regarding the absence of thyroid malignancy. The thyroid evaluation and long-term follow up (from the Danish Pathology Data Bank) showed no cases of thyroid malignancy in the group of patients presenting with normal/increased ITN tracer activity. This leads to the suggestion of a clinical work-up in patients with 18F-FDG-PET-positive ITN: perform thyroid scintigraphy—if no ITN hypofunction is present, consider benign; if hypofunction is present, perform ultrasonography and FNAC according to EU-TIRADS recommendations [20].

## 5. Conclusions

The frequency of thyroid malignancy in incidental thyroid FDG uptake in our study in Eastern Denmark was 11 patients of 279 patients included (3.9%); taking into consideration only the patients who received further thyroid evaluation, the frequency was 8.0%, and among all proceeding to surgery (or autopsy), the frequency was 50%.

The widespread increasing use of imaging studies leads to more incidentalomas being discovered, and as hospital resources are not unlimited—and to treat patients in an optimal manner—unnecessary biopsies and lobectomies/thyroidectomies should be avoided. Therefore, we suggest the use of a thyroid scintigraphy in the work-up of patients with incidental thyroid FDG uptake, as this diagnostic method can be used to select those patients who should undergo biopsy, leaving those with scintigraphically hyper- or normal-functioning nodules.

## Figures and Tables

**Figure 1 diagnostics-11-00557-f001:**
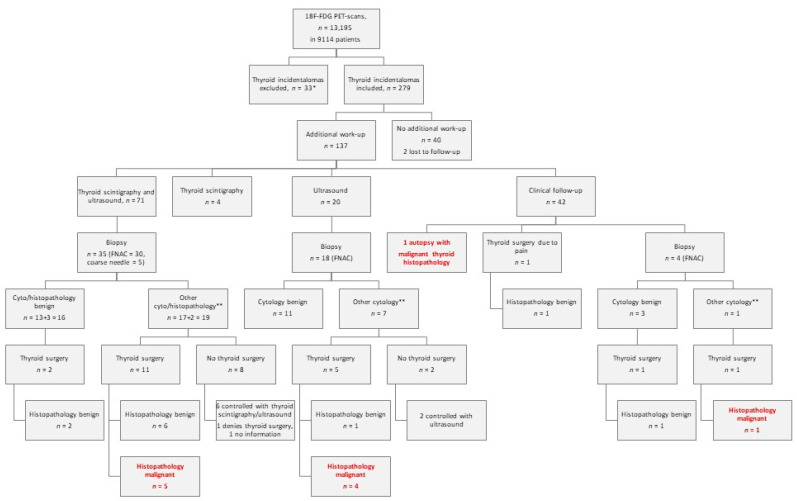
Flowchart of patients with thyroid incidentalomas. * Previous 18F-fluorodeoxyglucose positron emission tomography/computed tomography (18F-FDG-PET/CT) scan with thyroid uptake or known/suspicion of malignant thyroidal disease. ** Other cytology refers to malignant (Bethesda VI), suspicious for malignancy (Bethesda V), follicular neoplasia (Bethesda IV), atypical changes (Bethesda III), not representative/inconclusive (Bethesda I), or malignancy in coarse needle biopsies. All thyroid malignancies in surgical specimens are in this color.

**Figure 2 diagnostics-11-00557-f002:**
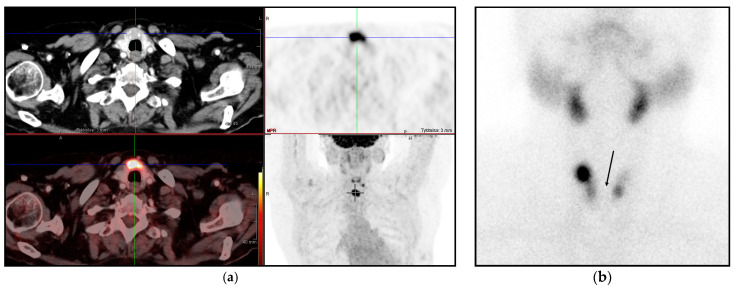
18F-FDG-PET/CT (**a**) displayed focally high FDG uptake in the isthmus of the thyroid, and (**b**) 99mTc-pertechnetate thyroid scintigraphy of the same patient showed a multinodular thyroid with no tracer uptake in the isthmus, a “cold” nodule (arrow). Subsequent surgery and pathology revealed a papillary carcinoma.

**Figure 3 diagnostics-11-00557-f003:**
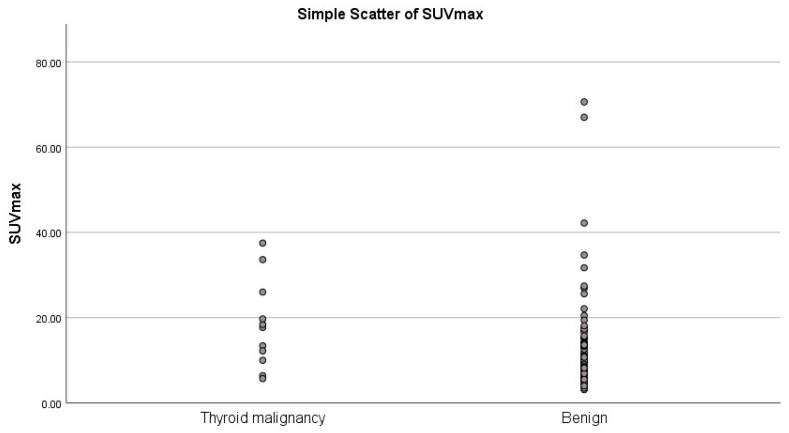
SUVmax in the group with thyroid malignancy (*n* = 11) vs. 125 cases with further thyroid evaluation and no assumed thyroid malignancies.

**Figure 4 diagnostics-11-00557-f004:**
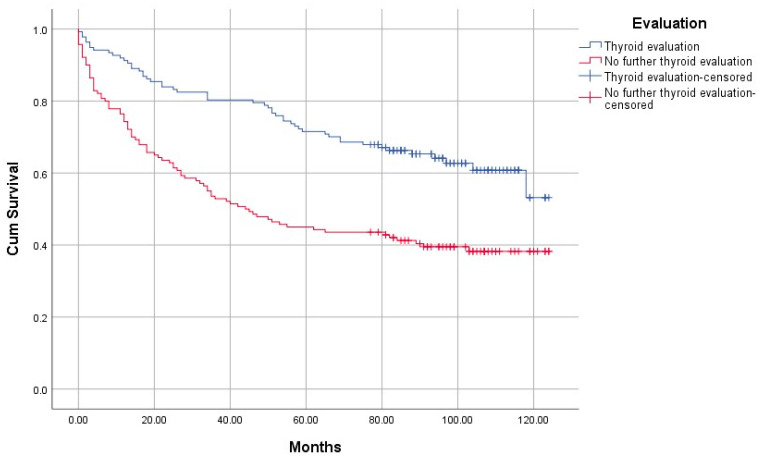
Survival curves of the two groups, the blue group received further thyroid evaluation (*n* = 137), while the red group did not (*n* = 140). In two patients, we have no information on eventual death/survival. Censored means end of follow-up time.

**Table 1 diagnostics-11-00557-t001:** Maximum standard 18F-fluorodeoxyglucose (FDG) uptake value (SUVmax), fine-needle aspiration cytology (FNAC), and plasma thyroid-stimulating hormone (P-TSH).

	Patients with Thyroid Evaluation (*n* = 137)		No Further Thyroid Evaluation (*n* = 142)	Statistics
	Thyroid Malignancy (*n* = 11)	No Proven Thyroid Malignancy (*n* = 126)		
Age at inclusion (mean, ± SD)	62 ± 15 years	60 ± 14 years	64 ± 14 years	*¤ p* = 0.016 # *p* = 0.591
Gender	8 females (73%), 3 males	95 females (76%), 31 males	113 females (79%), 29 males	
Thyroid scintigraphy	5 (45%)	70 (56%)	-	
Hypofunction	5	34		
Normal function	0	21		
Hyperfunction	0	13		
Inconclusive/missing information	0	2		
FDG uptake				
Focal	6	82	70	
Multifocal	4	23	25	
Diffuse	1	21	47	
SUVmax, median (interquartile range)	17.7 (10.0–26.0)	8.3 (6.1–13.0)	7.8 (5.9–10.9)	*¤ p* = 0.081 # *p* = 0.006
FNAC/coarse needle biopsy	10/0	42/5	-	
Surgery	10 (+1 autopsy)	11	-	
P-TSH (within ±30 days of index PET/CT)	*n* = 6	*n* = 50	-	
Low	2	5		
Normal	3	37		
Elevated	1	8		

¤ Groups with vs. without further evaluation. # Groups with vs. without malignancy (all with further thyroid evaluation).

## Data Availability

All relevant data can be found in the manuscript, and the remaining data (18F-FDG-PET/CT-scans, thyroid scintigraphies and patient reports etc.) cannot be made publicly available for ethical and legal reasons, as public availability would compromise patient confidentiality, and local (Danish) legislation prohibits public availability of the data. The study was approved by the Danish Data Protection Agency at Region Hovedstaden (RH-2015-227, I-Suite 04178) on 27 October 2015.

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
