# Peer review of "Long-Term Follow-Up of Thyroid Incidentalomas Visualized with 18F-Fluorodeoxyglucose Positron Emission Tomography—Impact of Thyroid Scintigraphy in the Diagnostic Work-Up"

_diagnostics, 2021, doi:10.3390/diagnostics11030557_

Round 1

Reviewer 1 Report

This is good info with population data. The final case number is a bit low and the result is ok.

The thyroid scintigraphy used Tc-99m, which has some limitation and need to be stress a bit further.

The flow chart is great.

We'd like to see a sample image of PET, and scintigraphy.

Reviewer 2 Report

The authors addressed some of my previous comments.

Although not very original, it is an interesting real-world description of actual clinical practice after finding incidental thyroid FDG-uptake in an (mostly) oncologic population. It shows us what decisions clinicians make and I think it is nice to show dat there still is a potential role for thyroid scintigraphy to exclude malignancies without performing a FNA. As it is less invasive, it might be an elegant first step of approach in these patients. In this paper however almost all scintigraphies are performed together with US and FNA is only performed in about half and it is not easy to see wheter or not it was US or scintigraphy that led to this discision.

Discussion is critic, presentation is average but most conclusions are not new or unexpected.
